# MIND: Material Interface Generation from UDFs for Non-Manifold Surface Reconstruction

**Xuhui Chen**[1,2], **Fei Hou**[1,2]*, **Wencheng Wang**[1,2]*, **Hong Qin**[3], **Ying He**[4]

[1]Key Laboratory of System Software (CAS) and State Key Laboratory of Computer Science, Institute of Software, Chinese Academy of Sciences
[2]University of Chinese Academy of Sciences
[3]Department of Computer Science, Stony Brook University
[4]College of Computing and Data Science, Nanyang Technological University
{chenxh, houfei, whn}@ios.ac.cn qin@cs.stonybrook.edu yhe@ntu.edu.sg

## Abstract

Unsigned distance fields (UDFs) are widely used in 3D deep learning due to their ability to represent shapes with arbitrary topology. While prior work has largely focused on learning UDFs from point clouds or multi-view images, extracting meshes from UDFs remains challenging, as the learned fields rarely attain exact zero distances. A common workaround is to reconstruct signed distance fields (SDFs) locally from UDFs to enable surface extraction via Marching Cubes. However, this often introduces topological artifacts such as holes or spurious components. Moreover, local SDFs are inherently incapable of representing non-manifold geometry, leading to complete failure in such cases. To address this gap, we propose MIND (Material Interface from Non-manifold Distance fields), a novel algorithm for generating material interfaces directly from UDFs, enabling non-manifold mesh extraction from a global perspective. The core of our method lies in deriving a meaningful spatial partitioning from the UDF, where the target surface emerges as the interface between distinct regions. We begin by computing a two-signed local field to distinguish the two sides of manifold patches, and then extend this to a multi-labeled global field capable of separating all sides of a non-manifold structure. By combining this multi-labeled field with the input UDF, we construct material interfaces that support non-manifold mesh extraction via a multi-labeled Marching Cubes algorithm. Extensive experiments on UDFs generated from diverse data sources, including point cloud reconstruction, multi-view reconstruction, and medial axis transforms, demonstrate that our approach robustly handles complex non-manifold surfaces and significantly outperforms existing methods. The source code is available at https://github.com/jjjkkyz/MIND.

## 1 Introduction

Signed Distance Fields (SDFs) are a widely adopted implicit representation for watertight surfaces due to their simplicity and effectiveness. The sign in SDFs clearly distinguishes the inside and outside of a surface, enabling straightforward surface extraction via well-established methods such as Marching Cubes (MC) [1]. While recent adaptations of SDF [2–5] incorporate constraints to support open surface reconstruction, they remain inadequate for capturing non-manifold structures.

Unsigned Distance Fields (UDFs), in contrast, eliminate the need for sign information and provide a more flexible framework capable of representing a wide range of surface topologies, including

---

*Corresponding authors.

39th Conference on Neural Information Processing Systems (NeurIPS 2025).

open and closed surfaces, non-manifold geometries, and shapes with complex internal structures [6–19]. However, this flexibility comes at a significant cost: the absence of sign information makes it significantly harder to identify zero-level sets, especially in the presence of non-manifold structures.

Several methods have been proposed for surface extraction from UDFs. A common strategy involves reconstructing local SDFs from UDFs using gradient-based estimation [6, 11, 20] or neural prediction [21] to approximate sign information and identify zero-level set intersections. While these methods benefit from the efficiency of Marching Cubes, they are highly sensitive to UDF inaccuracies, often resulting in holes and redundant components. Other approaches [22, 23] generalize dual contouring [24] to improve reconstruction quality, but they often introduce unintended non-manifold artifacts due to inconsistent topological handling. Mesh deformation methods, such as DCUDF [25, 26], improve robustness by iteratively shrinking an initial double-layered manifold surface to fit the target geometry. However, since the initial surface is always manifold and the deformation process preserves this structure, these methods are inherently incapable of capturing non-manifold geometries. Mesh extraction algorithms based on Dual Contouring [24] have the ability to extract non-manifold structures, but they also generate a large number of non-manifold faces in manifold regions. To our knowledge, there is currently no method that can effectively extract the correct non-manifold structures from UDFs.

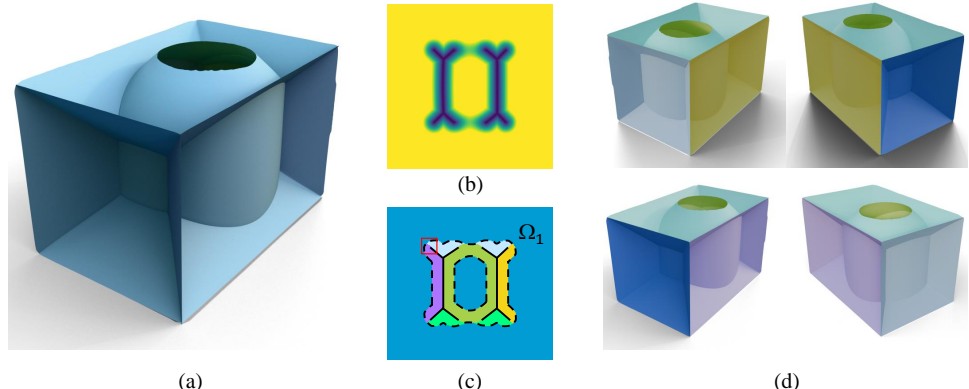

(a)          (b)          (c)          (d)

Figure 1: An open non-manifold surface (a) with 7 sides, including the top, bottom, left, right, front, back, and inner regions. Given the input unsigned distance field (A cross section is illustrated in (b)), we generate the corresponding material interface (c). For supporting open surfaces, we generate an envelope $\Omega_1$ (the dashed lines in (c)) enclosing the surface. To generate MI partitions, it needs to fill the gaps, e.g., the gap within the red box in (c), between $\Omega_1$ and the surface boundaries. We extend the surface boundaries slightly to intersect with $\Omega_1$. The redundant faces are removed while extracting the surface by M3C [27]. The reconstructed mesh in shown in four views (d) where each side is highlighted in a different color for clarity.

On the other hand, non-manifold structures are ubiquitous in many applications, such as anatomical modeling [28, 29], composite materials [30, 31], multi-phase fluids [32–35], and bubble simulations [36, 37]. These structures are characterized by complex topologies and often arise as interfaces between multiple materials, commonly referred to as **material interfaces** (MIs) [38]. An MI defines a partitioning of the spatial domain into multiple labeled regions $\{F_1, ..., F_n\}$ as shown in Figure 1(c).

Traditional MI representations are limited to closed surfaces, i.e., surfaces without boundaries. As illustrated in Figure 1, we define an enclosing envelope $\Omega_1$ around the surface. The surface boundaries are extended slightly to intersect with $\Omega_1$ to form MI partitions inside $\Omega_1$. The outside of $\Omega_1$ is treated as background and assigned the label $F_0$. This generalization allows us to deal with not only closed but also certain open non-manifold surfaces and to apply multi-label Marching Cubes methods, such as M3C [27], to reconstruct non-manifold meshes from such labeled partitions. Redundant surfaces, such as faces adjacent to the background, are removed.

However, MI is not a universal representation, as it requires predefined partition domain information. In practical applications, MIs are typically defined by known functions (e.g., from fluid simulations) or derived based on numerical priors (e.g., from CT images). In contrast, UDFs serve as a more universal

representation and have been widely adopted in many classic 3D reconstruction tasks, such as point cloud reconstruction [6–12], multi-view image reconstruction [13–16] and 3D generation [17–19].

To address these limitations, we introduce a novel algorithm to generate MIs from the input UDFs without requiring predefined partition domain information, which enables accurate non-manifold surface extraction from MIs.

Our method consists of three key steps. First, we generate a two-labeled field to distinguish between the two sides of a local surface patch using positive and negative signs. Second, we extend the local two-signed field into a global multi-labeled field, assigning unique labels to each side of a non-manifold surface. The multi-labeled field is then combined with the input UDF to generate the target MI. Third, we refine the extracted mesh from the MI to ensure both visual accuracy and topological coherence. We evaluate our method on a variety of datasets and UDF learning methods. Experimental results demonstrate that our approach generates clean meshes that accurately capture non-manifold structures, where existing methods often fail.

The main contributions of the paper are as follows:

1. We develop an algorithm for generating MIs from learned UDFs, enabling robust non-manifold surface reconstruction without requiring prior knowledge of MIs. By extending the definition of the MI, Our approach effectively handles both closed models and open models.

2. We introduce a novel algorithm that extends the local two-sided field—capable of distinguishing the two sides of local manifold patches—into a global multi-labeled field, enabling the differentiation of multiple sides of non-manifold surfaces.

3. We conduct extensive evaluations across diverse datasets and UDF learning methods. Experimental results demonstrate the capability of our method in extracting clean and accurate manifold and non-manifold meshes, outperforming existing techniques.

## 2 Related Works

### 2.1 Manifold Reconstruction

Recently, deep learning approaches have gained traction in surface reconstruction. These methods learn SDFs [39–49] or occupancy fields [50, 51] using neural networks directly from point clouds or multi-view images. These methods typically extract surfaces from signed distance fields [1, 52], which inherently guarantee watertight manifold models. Extensions of SDFs to support open surfaces typically involve introducing additional constraints or masks [2–5]. While these methods offer greater adaptability and flexibility in modeling, they remain fundamentally restricted to manifold surfaces.

### 2.2 Non-manifold Reconstructions

Unsigned distance fields have emerged as a promising alternative for representing surfaces with diverse topologies, including open surfaces, non-manifold geometries, and shapes with complex internal structures [6–11, 13–19, 53]. By discarding the sign term of SDFs, UDFs offer greater flexibility, enabling the representation of complex models, including non-manifold surfaces. However, most UDF-based methods primarily focus on open manifold surfaces, with limited exploration of non-manifold surface reconstruction. The primary obstacle lies in the lack of a robust mesh extraction algorithm tailored for non-manifold structures from UDFs.

Recent advancements have focused on extracting the zero-level sets from UDFs using modified Marching Cubes. Some methods [6, 11, 20, 21] reconstruct local sign information to determine edge intersections. DCUDF [25] refines the mesh of non-zero level sets to approximate the zero-level set through shrinking, producing double-layered results. However, these methods fail to effectively handle non-manifold geometries.

For non-manifold structures, sampling point clouds from UDFs and applying non-manifold-specific methods [54] have been explored but suffer from low accuracy and robustness. Dual Contouring-based methods, such as NDC [22] and DMUDF [23], have the potential to generate non-manifold geometries but either lack generalizability or introduce unintended non-manifold artifacts. Manifold DC [55] avoids such artifacts but cannot model non-manifold structures.

Existing non-manifold mesh extraction algorithms are predominantly applied in the context of material interfaces (MIs). MI represents a collection of regions where the target surface corresponds to the intersection of different regions. MI is widely utilized in partitioned domains, such as anatomical structures [28, 29], composite materials [30, 31], bubbles [36, 37], and multi-phase fluids [32–35], where space is naturally segmented into distinct regions.Using the MI as input, multi-label algorithms, such as M3C [27], handle non-manifold surfaces by leveraging explicit material interface definitions. While effective, they require predefined region labels or arrangements, limiting their applicability.

In this paper, we aim to address the limitations of these approaches by introducing a novel method for generating MIs directly from UDFs. Unlike previous methods, our approach does not require prior knowledge of material interfaces or region labels, enabling the robust extraction of non-manifold surfaces directly from UDFs.

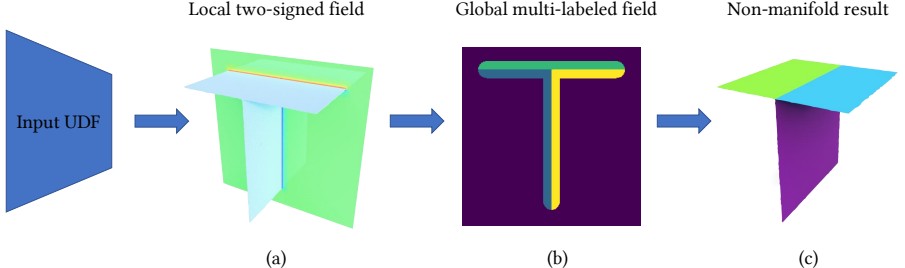

Figure 2: Pipeline: Starting with a learned UDF, we first sample a point cloud to compute a local two-signed field to differentiate the two sides of local manifold patches (a). We do not calculate regions far from the target face and label them as background (the green region in (a)). This is followed by generating a global multi-labeled field based on the two-signed field, which distinguishes all sides of the non-manifold surface (b). Finally, the non-manifold surface is extracted from the multi-labeled distance field using a multi-label MC algorithm (c).

## 3 Method

In this work, we generate MIs from input UDFs to enable the extraction of non-manifold meshes from UDFs. As illustrated in Figure 2, our method consists of three key steps: In Section 3.1, we construct a local two-signed field to distinguish the two sides of manifold patches within the input UDF. In Section 3.2, the local two-signed field is extended to a global multi-labeled field to capture the sides of non-manifold surfaces, forming the target MI. Finally, in Section 3.3, we describe how to extract non-manifold surface meshes from the MI.

### 3.1 Local Two-Signed Fields

To generate the MI of the input UDF, we need to segment the 3D space into different partitions. As shown in Figure 2, our first step is to generate a two-signed local side field that distinguishes the two sides of local manifold patches. Similar to the generalized winding number [56, 57], on a surface $S$, given consistently oriented normals $\mathbf{n_x}$ of points $\mathbf{x} \in S$, we introduce the following indicator function $w_S^l(\mathbf{q})$ to compute a side field for a query point $\mathbf{q}$:

$$w_S^l(\mathbf{q}) = \int_{\mathbf{x} \in \mathcal{N}_S(\mathbf{q})} \frac{(\mathbf{x} - \mathbf{q}) \cdot \mathbf{n_x}}{\|\mathbf{x} - \mathbf{q}\|^3 + \epsilon} d\mathbf{x}, \tag{1}$$

where $\epsilon$ is a small positive to avoid division by zero. Different from the generalized winding number, the region of integration is modified from the entire surface $S$ to a local neighborhood $\mathcal{N}_S(\mathbf{q})$ on $S$ around the point $\mathbf{q}$. This adjustment allows $w_S^l$ to distinguish between the two sides of a local manifold using positive and negative signs.

**Implementation Details** We extract a point cloud from the given UDF. Points are sampled randomly in space $\Omega_1$ where the UDF values are equal to a threshold $r_1$ and these points are projected toward local minima, similar to the approach in [10]. The point cloud is then downsampled using uniform grid voxels to obtain a uniform initial point cloud $\mathcal{P}$. Next, we apply [58] to compute oriented

normals for the points. Although [58] may result in flipped normals, the orientations are piecewise consistent, ensuring that most are oriented consistently. The discrete form of $w_S^l(\mathbf{q})$ is,

$$w_S^l(\mathbf{q}) = \sum_{\mathbf{x}_i \in \mathcal{N}_{\mathcal{P}}(\mathbf{q})} \frac{(\mathbf{x}_i - \mathbf{q}) \cdot \mathbf{n}_i}{\|\mathbf{x}_i - \mathbf{q}\|^3 + \epsilon}. \tag{2}$$

We discretize the space into voxels and compute the side field only for the voxels $o_i$ inside $\Omega_1$, as points far from the surface are not of interest. Any voxel outside $\Omega_1$ is considered background and assigned the label $F_0$. $w_S^l(\mathbf{q})$ is locally defined and is able to distinguish the two sides where the normals are properly defined. In particular, in regions near non-manifold edges or regions with flipped normals, $w_S^l$ is not well-defined. These issues will be refined in the following Section 3.2.

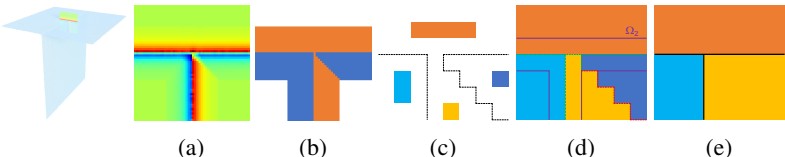

|  (a)  |  (b)  |  (c)  |  (d)  |  (e)  |

Figure 3: Illustration of global multi-labeled field generation from the local two-signed field on a T-shaped model. The close-up view of the cross-section on the non-manifold structure is provided. The local two-signed field $w_S^l$ is first computed (a). Applying connected component labeling to the local two-signed field introduces artifacts due to small "tubes" (b). Erosion effectively removes these connected "tubes" (c). We solve the Equation 3 to fill the blank region (d). Comparing to dilate operation, it produce a more consistent boundary to the origin labeling (dash line). But our current result is over-segmented. We introduce an envelope $\Omega_2$ that is closer to the target surface than $\Omega_1$. As shown in (e), the partition boundaries inside $\Omega_2$ is shown in green and outside in red. We merge two regions whose most adjacent boundaries are in red to get the final labeling result (e).

## 3.2   Global Multi-Labeled Fields

In this section, we generate the global multi-labeled distance field, which is able to distinguish all sides of a non-manifold surface, from the local two-signed side field. We cluster voxels based on the two signs of $w_S^l$ by applying the 3D connected component labeling algorithm[2]. This algorithm assigns a label $F_i$ to each voxel that is connected and has the same sign (positive or negative). As a result, voxels within $\Omega_1$ are split into a set of partitions $\{R_k\}$. However, non-manifold or normal flipping regions, where three or more partitions coincide, are scarcely possible to be properly divided only by the two signs of $w_S^l$. As shown in Figure 3, a partition may span across non-manifold edges via a narrow "tube". Since this tube is thin, a simple morphological erosion can remove it, causing the remaining voxels of the partition to become disconnected.

We denote the eroded voxels by $R_k^e$ and the remaining voxels by $R_k^r$. For the voxels in all the connected components of $R_k^r$, we assign different new labels $F_k$ to different connected components of $R_k^r$. Each voxel in $R_k^e$ should be labeled by one of the labels of $F_k$. The goal is to minimize variations in neighboring labels of $R_k^e$. Therefore, we minimize the following energy for labeling:

$$\min_f \sum_{o_i \in \cup_k R_k^e} D\left(f(o_i)\right) + \sum_{(o_i, o_j) \in \mathcal{N}} V\left(f(o_i), f(o_j)\right),$$
$$s.t.\ f(o_i) = f_s(o_i),\ o_i \in \cup_k R_k^r$$
$$D\left(f(o_i)\right) = \begin{cases} 0, & \text{if } f(o_i) \in F_k \text{ or } F_k = \Phi \\ 1, & \text{otherwise} \end{cases},\ (o_i \in R_k^e) \tag{3}$$
$$V\left(f(o_i), f(o_j)\right) = \begin{cases} 0, & \text{if } f(o_i) = f(o_j) \\ 1, & \text{otherwise} \end{cases}$$

Here, $f(o_i)$ represents the to be solved label assigned to voxel $o_i$, $f_s(o_i)$ denotes the label assigned to voxels in $R_k^r$, which are fixed, and $\mathcal{N}$ refers to the set of neighboring voxels. The function $V(\cdot)$ minimizes label changes, while $D(\cdot)$ ensures that most of the interfaces between partitions remain

---

[2]`https://github.com/seung-lab/connected-components-3d`

invariant. Occasionally, $R_k$ may be so small that $R_k^r$ and $F_k$ are empty sets. In such cases, we omit the constraint. Equation (3) can be solved using $\alpha$-expansion [59].

After refinement, the partitions ensure that the two sides of a non-manifold belong to different partitions. However, over-segmented partitions may exist, as shown in Figure 3, which are unavoidable in the two-signed field near non-manifold edges. These over-partitions should be merged. Otherwise they would lead to redundant surfaces in the reconstructed mesh. Two partitions that are not separated by the surface $S$ should be merged. Directly assessing this condition can be tricky, so instead, we construct another envelope, $\Omega_2$, by extracting an iso-surface at value $r_2$ ($r_2 < r_1$) from the UDF. This gives us the relation $S \subset \Omega_2 \subset \Omega_1$. If two partitions are separated by $S$, their boundary voxels should lie within $\Omega_2$. Conversely, if two partitions are not separated by $S$, most of their boundary voxels should be outside $\Omega_2$, but still within $\Omega_1 - \Omega_2$. As illustrated in Figure 3, through these simple tests, we can effectively merge redundant partitions, ensuring that different sides of a non-manifold surface belong to different partitions, and no further merging of partitions is needed. These partition labels along with the input UDF constitute the target MI.

### 3.3 Non-Manifold Surface Extraction

With the target MI, we can extract the non-manifold mesh $\mathcal{M}$ using a multi-label Marching Cubes algorithm. Specifically, we adopt the M3C method [27] with minor modifications. Instead of interpolating at the midpoint of each cube edge, we use the value of $w_S^l$ to determine the intersection points, enhancing accuracy. We do not generate the face associated with the background label $F_0$ because there is no target surface at the interface between the background and other regions, which also enables open model reconstruction. To further refine the mesh, redundant triangular faces extending from surface boundaries outside $\Omega_2$ are removed.

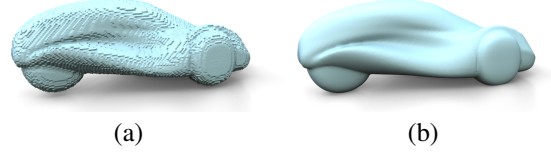

(a)                    (b)

Figure 4: The Multi-Labeled Field computed from the point cloud, while having the correct topology, generates a noisy mesh because its zero level set is misaligned with the target surface (a). We use the input UDF to refine the result (b).

However, the accuracy of $w_S^l$ remains limited by discretization. To address this limitation, we refine the extracted mesh $\mathcal{M}$ by optimizing its alignment with the input UDF as shown in Figure 4. Inspired by DCUDF [25], we fine-tune $\mathcal{M}$ by minimizing its UDF values for improved accuracy while incorporating a Laplacian regularization term to maintain the mesh's shape and prevent face folding. Unlike DCUDF, our mesh $\mathcal{M}$ contains non-manifold edges, where traditional Laplacian computation is not well-defined. For a point $\mathbf{p}_m$ on a non-manifold edge, using all adjacent points to compute the Laplacian fails to prevent face folding, as illustrated in Figure 5. To overcome this, we group adjacent triangular faces based on the labels they border. Every triangular face belongs to two groups. Each group of triangular faces forms a manifold mesh. For a point $\mathbf{p}_i \in \mathcal{M}$, the Laplacians are computed separately in each group of adjacent faces. For example, for a point on the non-manifold edge of a T-shape, there are 3 groups of adjacent faces and 3 Laplacian terms. We optimize the following loss function to refine the mesh,

$$\min_\pi \sum_{s \in \mathcal{S}} \left( \sum_{\mathbf{p}_i \in \mathcal{M}^s} f\big(\pi(\mathbf{p}_i)\big) + \lambda_1 \sum_{\mathbf{p}_i \in \mathcal{M}^s} \left\| \pi(\mathbf{p}_i) - \frac{1}{|\mathcal{N}^s(\mathbf{p}_i)|} \sum_{\mathbf{p}_j \in \mathcal{N}^s(\mathbf{p}_i)} \pi(\mathbf{p}_j) \right\|^2 \right), \qquad (4)$$

where $f(\cdot)$ denotes the UDF values and $\pi(\mathbf{p}_i)$ is the new location of point $\mathbf{p}_i$ after optimization. $\mathcal{S}$ denotes the set of signs and $\mathcal{M}^s$ is the sub-mesh whose faces border on the sign $s$. $\mathcal{N}^s(\mathbf{p}_i)$ denotes the 1-ring neighboring points of $\mathbf{p}_i$ in $\mathcal{M}^s$. The first term, $f(\pi_1(\mathbf{p}_i))$, drives the points $\mathbf{p}_i$ toward the local minima of the UDF. The second Laplacian term prevents the triangular face from folding.

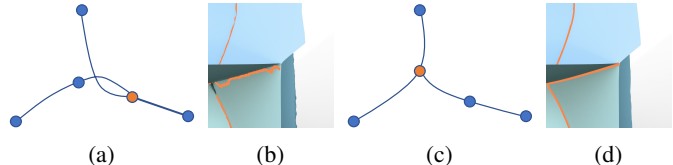

(a)            (b)            (c)            (d)

Figure 5: Laplacian constraint of non-manifold edges. For a point (orange) on a non-manifold edge, traditional Laplacian constraint fails to prevent adjacent faces from folding. By computing the Laplacian loss within each labeled region separately, our method effectively avoids self-intersections of the surface.

## 4 Experimental Results

### 4.1 Experimental Setup and Hyperparameters

We normalize 3D models to fit within $[-0.5, 0.5]^3$ and use a bounding box of $[-0.6, 0.6]^3$ to contain the UDFs. For calculating the local two-signed field, we sample 1 million points on the $r_1$ level set and optimize their positions to align with the local minima of the UDF. The resolution is set to $256^3$, resulting in a voxel size of 0.0046. The voxel size for point cloud downsampling[3] is set to 0.005, which is slightly higher than 0.0046. While the downsampled point cloud has a density of 0.005, we set $r_2 = 0.01$ to generate a continuous $\Omega_2$. To ensure $\Omega_1$ is larger than $\Omega_2$, $r_1$ is set to 0.05. We erode the local two-signed field 2 times before generating the global multi-labeled field. Two partitions in the global multi-labeled field are merged if the number of boundary voxels within $\Omega_1 - \Omega_2$ is three times greater than the number of voxels within $\Omega_2$. A more detailed experiment of hyperparameters can be found in Section A of appendix.

We use M3C [27] to extract meshes, implemented in Dream3D[4]. We then optimize the output of M3C with Equation 4 for 200 iterations, using a Laplacian weight of 1000, to generate the final result. Although several hyperparameters are introduced in our paper, most of them correspond to the resolution and exhibit generalizability across different types of learned distance fields. All results are tested on a single NVIDIA V100 GPU.

### 4.2 Comparisons

**Baselines** To the best of our knowledge, no prior work investigates generating MIs from UDFs. Since the extracted meshes depend on MI qualities, we compare our method against two unsupervised UDFs mesh extraction algorithms, including DCUDF [25], and DMUDF [23]. While DCUDF uses double-layered manifold meshes to approximate non-manifold structures, DMUDF is capable of generating non-manifold edges. To assess the topological correctness of the extracted meshes, we compute geodesic distances, which are highly sensitive to topological features. Specifically, we use the heat method [60] with a non-manifold Laplacian [61] applied to the extracted meshes. For comparison, geodesic distances are also computed on dense points, serving as a reference.

**UDFs Learned from Point Clouds** We learn UDFs from unoriented point clouds using CA-PUDF [6], LevelSetUDF [7], and DEUDF [8], respectively. The results are shown in Figure 6, with non-manifold edges highlighted in red. We compare our method with DMUDF and DCUDF.

DMUDF [23], a Dual Contouring variant, is capable of generating non-manifold edges. However, the process of generating non-manifolds in DC is often uncontrolled, resulting in a significant number of non-manifold edges appearing in regions that should remain manifold. Furthermore, DMUDF utilizes an octree structure to accelerate the algorithm. To determine whether the target surface exists within a specific leaf node, DMUDF relies on a key criterion based on the UDF value at the node's center point. This approach is highly sensitive to UDF accuracy, making it prone to failure in the presence of noise or inaccuracies in the UDF. Such issues can cause the octree subdivision process to terminate prematurely. As highlighted in DUDF [9], most UDF learning methods prioritize accurate distance predictions near the target surface but neglect accuracy in regions farther away. This limitation can

---

[3]https://www.open3d.org
[4]https://github.com/BlueQuartzSoftware/DREAM3D

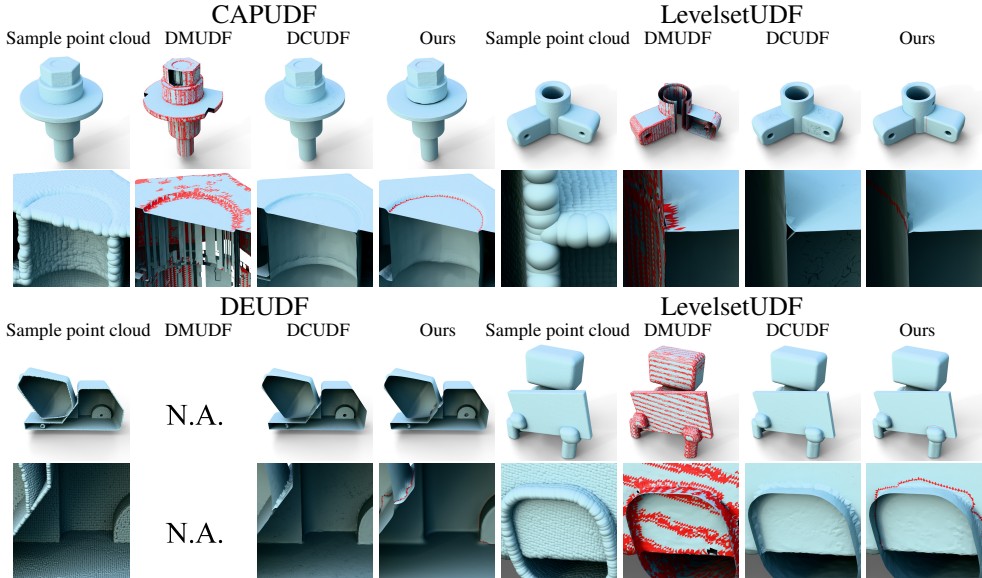

Figure 6: Non-manifold surface extraction from UDFs learned by CAPUDF [6], LevelSetUDF [7], and DEUDF [8]. We present the sampled point cloud on the UDFs as the GT (Ground Truth). We compare our surface extraction method with DCUDF and DMUDF, highlighting non-manifold edges in red in the results. While DMUDF frequently produces non-manifold edges in regions that should be manifold, DCUDF consistently generates double-layered manifold meshes, leading to a failure in preserving the correct topology of the target surfaces.

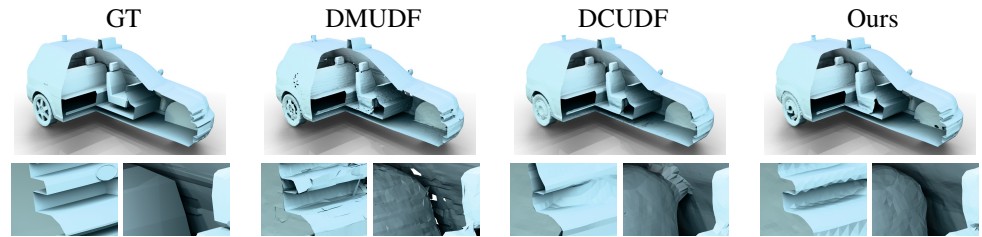

Figure 7: Surface extraction from UDFs learned by CAPUDF [6] on the ShapeNet-Car dataset [62].

result in significant missing regions in DMUDF's output when a node's center point is far from the vicinity of the target surface. DCUDF [25] approximates the target non-manifold surfaces using a double-layer mesh. Although the results visually align with the target surface, the lack of exact coincidence between the two layers often introduces undesired artifacts, such as redundant shadows, during rendering.

To assess the quality of the extracted meshes, we sample 100K points on the mesh and employ the Chamfer distance L2 as a geometrical metric. Table 1 presents the results on the ShapeNet-Car [62] dataset and DeepFashion3D [63] dataset. The visualization results of ShapeNet-Car dataset are shown in Figure 7, where our visual results are the best.

Table 1: Evaluation on the ShapeNet-Car [62] dataset learned from CAP-UDF[6] and DeepFashion3D dataset [63] learned from DCUDF[25].

| Dataset | DMUDF | | | DCUDF | | | Ours | | |
|---|---|---|---|---|---|---|---|---|---|
| | Mean | Median | Std | Mean | Median | Std | Mean | Median | Std |
| ShapeNet-Car | 3.086 | 2.564 | 2.112 | 3.290 | 2.770 | 2.252 | 2.917 | 2.447 | 2.107 |
| DeepFashion3D | 1.792 | 1.383 | 0.512 | 1.825 | 1.495 | 0.535 | 1.770 | 1.339 | 0.507 |

**UDFs Learned from Multi-view Images**   Extracting surfaces from UDFs learned via multi-view images presents significant challenges, especially in scenes involving transparent objects or thin structures, where non-manifold surfaces are prevalent.

We evaluate our method and compare it with baselines on UDFs generated by NU-NeRF [43] on multi-view images of transparent objects. NU-NeRF learns two separate SDFs, corresponding to the outer and inner objects, and extracts their respective meshes using Marching Cubes. To prevent the inner SDF from producing meshes in the outer region, NU-NeRF uses the outer SDF as a mask during extraction. However, this masking approach often leads to redundant components forming along the mask boundaries. Although NU-NeRF employs a post-processing step to remove these redundant components, it creates discontinuities between the inner and outer meshes.

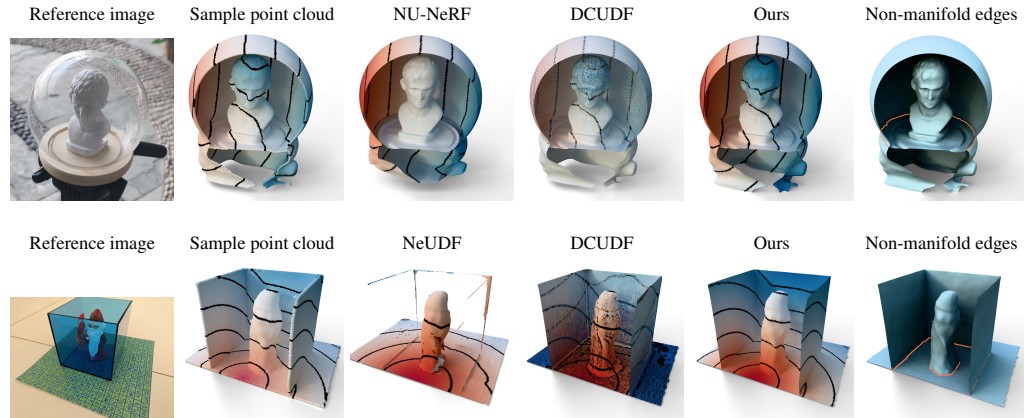

Reference image | Sample point cloud | NU-NeRF | DCUDF | Ours | Non-manifold edges

Reference image | Sample point cloud | NeUDF | DCUDF | Ours | Non-manifold edges

Figure 8: Non-manifold surface extraction from UDFs learned by NU-NeRF [43] (top) and NeUDF [15] (bottom). Geodesic distances computed on the extracted meshes are visualized to validate their non-manifold topology. The geodesic distances computed on sampled point clouds are used as reference for comparison. All baseline methods fail to accurately preserve the non-manifold structures in the extracted meshes.

To address this, we combine the two SDFs[5] to a single UDF for mesh extraction. The outer SDF value is directly applied in the outer region, while for the inner region, we use the minimum absolute value of the two SDFs. As shown in Figure 8, adopting the geodesic distance measure, we confirm that our results preserve the correct topology. For comparison, we also use DCUDF to extract the target surface. While DCUDF produces visually pleasing results, its double-layered structure prevents geodesic distances from diffusing between layers, highlighting its limitation in preserving topological consistency.

We also adopt NeUDF [15] for learning UDFs from multi-view images of transparent objects. The results are presented in Figure 8. NeUDF employs MeshUDF [20], a gradient-based Marching Cubes, for mesh extraction from learned UDFs. However, this approach fails in non-manifold regions due to the lack of a suitable lookup table for non-manifolds in standard Marching Cubes. Furthermore, unlike opaque objects whose boundary surfaces align with zero-level sets, transparent objects typically exhibit iso-values that are not close to zero, leading to complete reconstruction failure for transparent objects. DCUDF addresses this limitation by extracting non-zero level sets, allowing it to capture transparent objects. However, its double-layered mesh structure significantly compromises topological accuracy. In contrast, our method successfully reconstructs transparent objects and accurately models their non-manifold surfaces. This highlights the robustness and versatility of our approach compared to existing methods.

**UDFs Induced from Q-MDF [64]**   Non-manifold structures frequently appear in medial axes. Q-MDF [64] computes medial axes for watertight models through the joint learning of signed distance fields and medial fields (MF) [65]. It has been shown that the difference between the SDF and MF yields an unsigned distance field [64]. In the original Q-MDF pipeline, medial axes are extracted using DCUDF [25], which, as mentioned above, generates a double-layered manifold mesh. Consequently,

---

[5]We use the SDFs provided directly by the NU-NeRF authors.

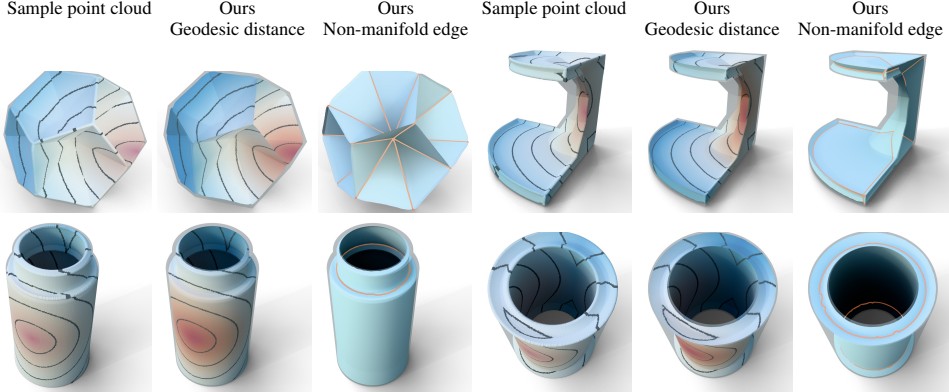

| Sample point cloud | Ours Geodesic distance | Ours Non-manifold edge | Sample point cloud | Ours Geodesic distance | Ours Non-manifold edge |

Figure 9: Non-manifold surface extraction from UDFs learned by Q-MDF [64]. The target surfaces are medial axes, characterized by numerous non-manifold structures. For clarity, we render both the watertight surfaces and their corresponding medial axes. Geodesic distances are computed on the extracted non-manifold medial axes and compared with those on the sampled point clouds for validation.

this approach fails to preserve the non-manifold characteristics of medial axes. By utilizing MIND, we extract high-quality, single-layered non-manifold medial axes, as demonstrated in Figure 9.

**Limitations** We use $\alpha$-expansion [59] to label voxel grids, which is time-consuming. The primary bottleneck arises from evaluating all potential label distributions across the entire voxel grid for each candidate label. A feasible acceleration strategy involves partitioning the voxel grid into blocks, where block-wise $\alpha$-expansion computation effectively reduces costs. Another alternative is to use dilation instead of alpha expansion, which has a time cost independent of the number of labels but yields an approximate solution. On the other hand, our approach requires an erosion operation, which may inadvertently remove small or thin MI regions and lead to missing facets, as shown in Figure 10.

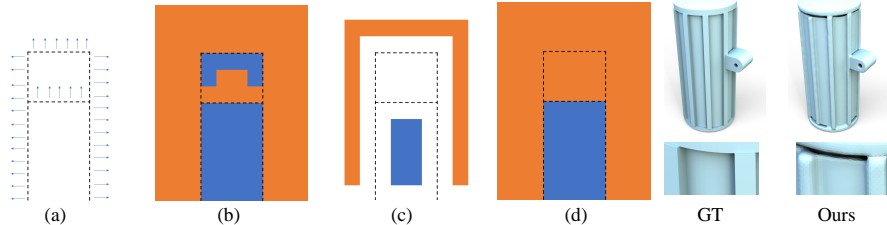

|      (a)      |      (b)      |      (c)      |      (d)      |      GT      |     Ours     |

Figure 10: For a small MI region (upper part) (a), the inner voxels of its local two-signed field (b) will be removed during the erosion process, resulting in no seed region in the entire internal space (c). Consequently, the structure cannot be recovered after $\alpha$-expansion (d).

## 5 Conclusions

In this paper, we introduce MIND, a novel algorithm to extract MIs from UDFs for non-manifold mesh extraction. By combining the strengths of material interfaces and unsigned distance fields, MIND supports non-manifold reconstruction from UDFs. MIND does not require pre-defined partition information, making it suitable for a broader range of scenarios. Our experimental results across various types of UDFs demonstrate the effectiveness of MIND in generating MIs for accurate non-manifold mesh reconstruction. In its current form, our implementation generates MI from pre-learned UDFs, making the presented method primarily a zero-level set extraction algorithm. It is highly desired to develop techniques that learn an MI directly from raw input data, such as point clouds or multi-view images. Such advancements could significantly broaden the range of applications for MI and enhance its utility in tackling complex reconstruction tasks.

## Acknowledgments and Disclosure of Funding

This project was partially supported by the National Key R&D Program of China (2023YFB3002901), the Research Projects of ISCAS (ISCAS-JCMS-202303, ISCAS-ZD-202401, ISCAS-JCZD-202402, ISCAS-JCMS-202403), and the Ministry of Education, Singapore, under its Academic Research Fund Grant (RT19/22).

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

## A  Hyper-parameter study

In our pipeline, we use a resolution-dependent voxel size to downsample the point cloud. As Figure 11 shows, a smaller voxel size does not affect the result but introduces a larger computational overhead. Conversely, if the voxel size is set too large (e.g., 0.01), the point cloud becomes too sparse and holes appear in the reconstructed model. Our $r_1$ and $r_2$ values are also set according to the resolution to maintain them within proper ranges. As shown in Figure 12, moderate changes in these parameters are acceptable, but excessive adjustments can cause problems. Though a large $r_1$ does not introduce issues beyond computational overhead, an overly small $r_1$ (e.g., 0.03) could cause the erosion step to delete all regions, leading to reconstruction failure. An excessively large $r_2$ (e.g., 0.02) can lead to insufficient region merging, whereas an overly small $r_2$ (e.g., 0.0025) can result in unnecessary merging.

| 0.002 | 0.005 | 0.0075 | 0.01 |

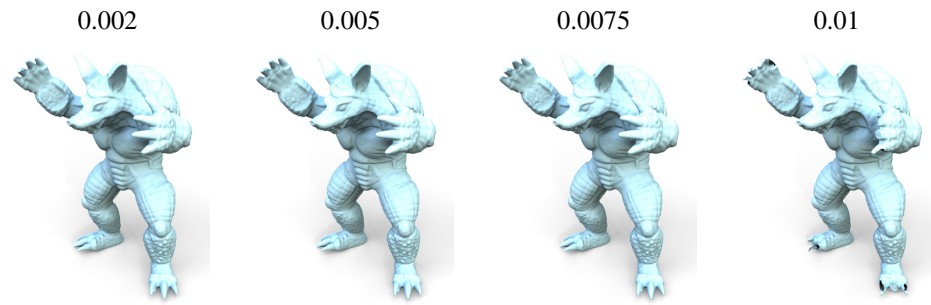

Figure 11: The reconstruction results with different downsampling voxel size.

| $r_1 = 0.1$ | $r_1 = 0.075$ | $r_1 = 0.05$ | $r_1 = 0.04$ | $r_1 = 0.03$ |

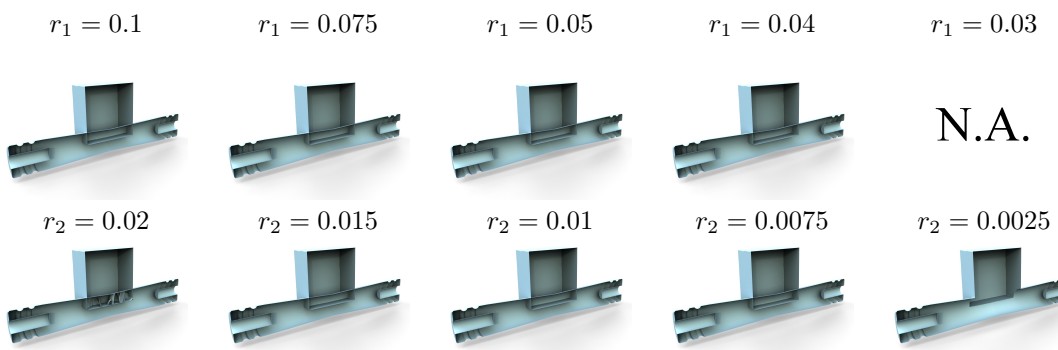

| $r_2 = 0.02$ | $r_2 = 0.015$ | $r_2 = 0.01$ | $r_2 = 0.0075$ | $r_2 = 0.0025$ |

Figure 12: Mesh reconstruction with different $r_1$ and $r_2$. N.A. means reconstruction failed.

## B  More Results

We present additional results in the appendix, including point cloud reconstruction, multi-view reconstruction, and medial axis transforms. As illustrated in Figure. 13, our method can model real-world non-manifold structures defined by multi-view images, such as intersections between transparent objects and overlapping thin plate structures. These configurations represent challenging cases for signed distance fields to represent in practical scenarios. For point cloud reconstruction and medial axis transformations, we present additional results in Figure. 14 and Figure. 15, demonstrating the versatility of our approach. We also collected some more complex data to further illustrate the robustness of our method, as shown in Figure 16.

## C  Failed Case

As illustrated in Figure 17, the two sides of the non-orientable surface are indistinguishable, resulting in the absence of a well-defined MI. Consequently, our method fails to generate MIs for the surface, leading to artifacts in the extracted mesh.

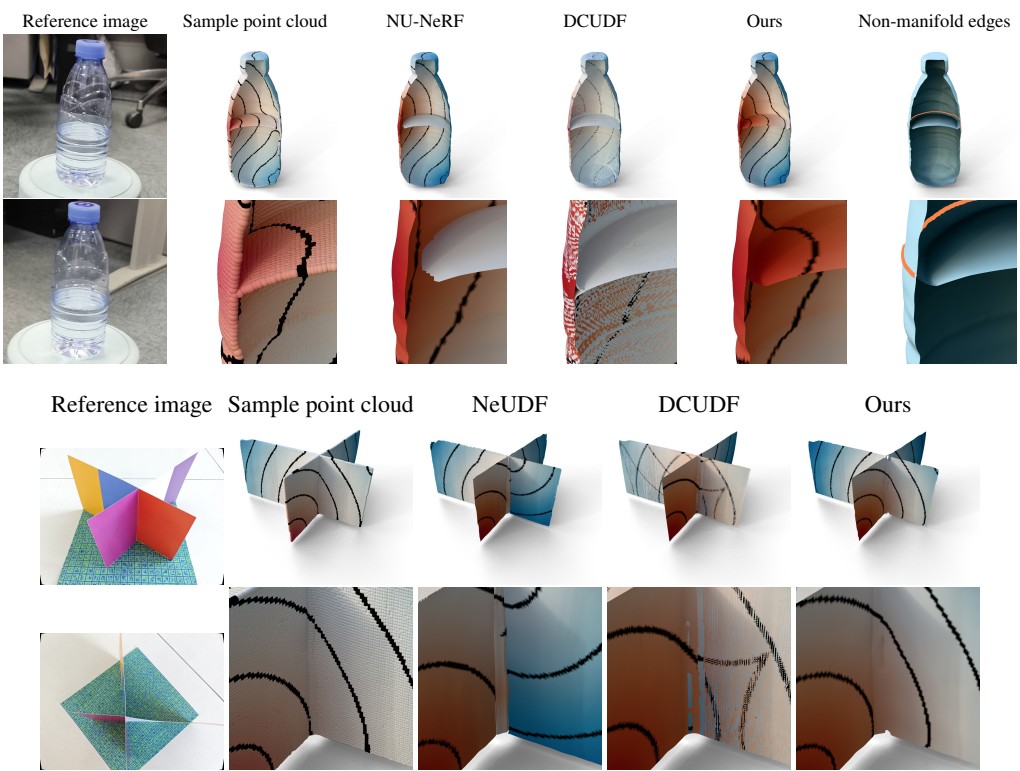

Figure 13: More results of Non-manifold surface extraction from SDFs/UDFs learned by NU-NeRF (top) and NeUDF (bottom). We use the SDFs provided directly by the NU-NeRF authors and convert them to UDFs.

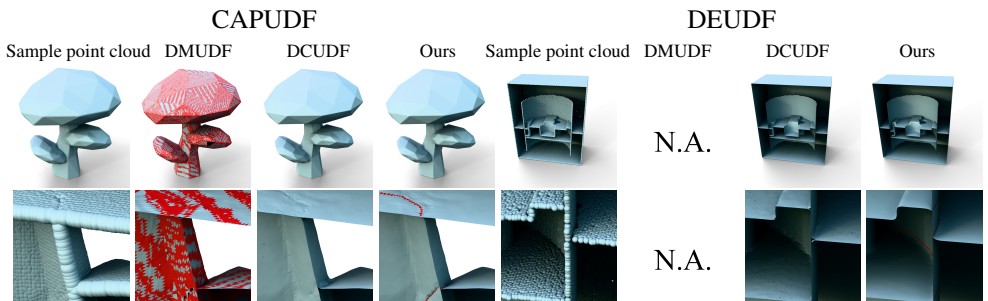

Figure 14: More results of non-manifold surface extraction from CAPUDF and DEUDF.

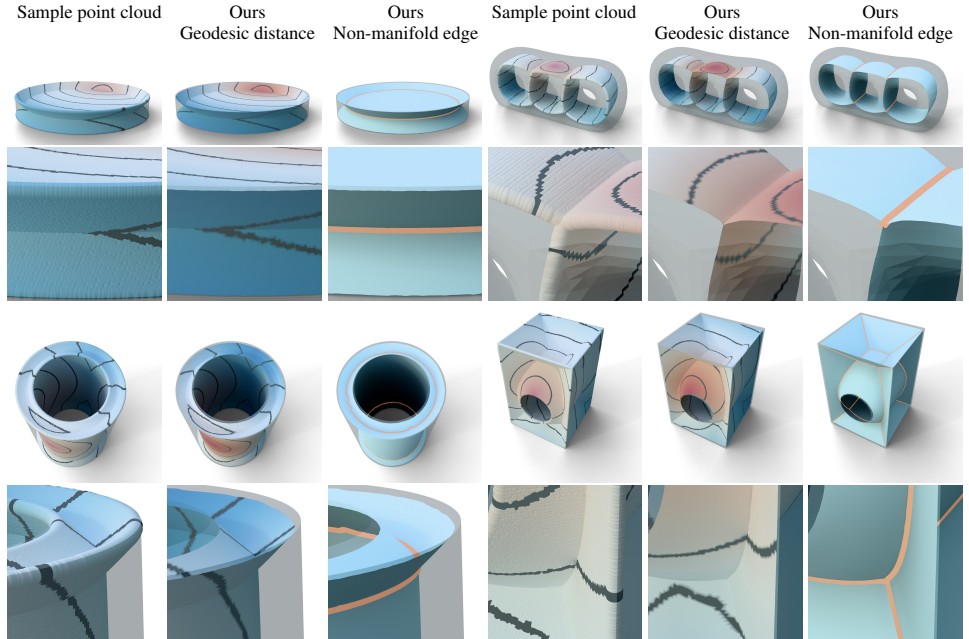

| Sample point cloud | Ours
Geodesic distance | Ours
Non-manifold edge | Sample point cloud | Ours
Geodesic distance | Ours
Non-manifold edge |

Figure 15: More results of non-manifold surface extraction from UDFs learned by Q-MDF [64].

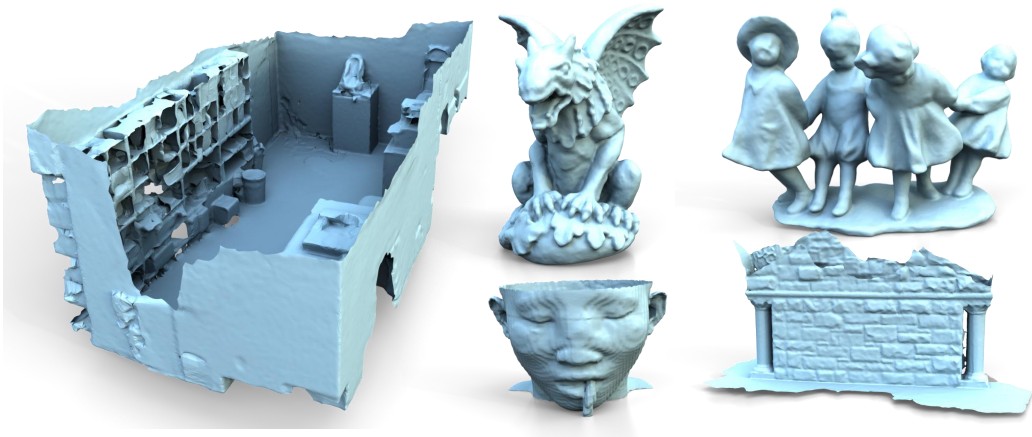

Figure 16: Results on more data, where the UDFs are learned by DEUDF [8]. We demonstrate the reconstruction results of indoor data, medical images, objects, and high fidelity surface data, which illustrates the applicability of our method.

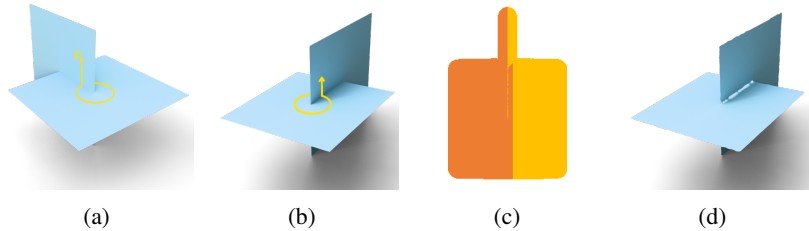

|  (a)  |  (b)  |  (c)  |  (d)  |

Figure 17: Non-orientable surfaces do not have proper MI definitions. (a) and (b) illustrate a non-orientable non-manifold surface where a walker can traverse from one side of a point to the opposite side without crossing a boundary. In such case, the multi-labeled field is undefined and thus fails to generate (c), from which the non-manifold surfaces cannot be extracted properly (d).

