# OpenReview forum: "MIND: Material Interface Generation from UDFs for Non-Manifold Surface Reconstruction"
_NeurIPS.cc/2025/Conference — NeurIPS 2025 poster_

### Official Review · Reviewer_Tx5y · 2025-07-02

**Clarity:** 3
**Significance:** 3
**Originality:** 3
**Rating:** 3
**Confidence:** 3

**Summary:**

To extract meshes from Unsigned distance fields (UDFs), the authors proposed a novel algorithm, MIND, to generate material interfaces directly from UDFs, enabling non-manifold mesh extraction from a global perspective. The effectiveness of MIND is demonstrated on different data sources including UDFs from point cloud reconstruction, multi-view reconstruction, and medial axis transforms.

**Questions:**

1.	What is the N.A. in Figure 5?
2.	In addition to chamfer distance, are there any other metrics to demonstrate the superiority of MIND?
3.	The presented results are obtained on simple scenarios. Can the non-manifold be applied to more realistic and complex scenarios, such as medical image scenarios?

**Ethical Concerns:**

["NO or VERY MINOR ethics concerns only"]

**Final Justification:**

I believe the results of the standard deviation of CD and a statistical significance analysis are important to evaluate if there is a significant improvement in the performance. However, I didn't see additional results on them.

**Limitations:**

The potential applications of the proposed method should be discussed in more detail.

**Paper Formatting Concerns:**

P4, "query point q,:"->"query point q:"

**Quality:**

3

**Strengths And Weaknesses:**

Strength:
The authors proposed MIND to extract no-manifold meshes directly from the UDFs. As they claimed, no previous method has effectively extracted non-manifold surfaces directly from UDFs.

Weakness:
1. The advantage of non-manifold surfaces compared to manifold surfaces should be discussed in more detail, including more realistic scenarios where manifold surfaces cannot work while non-manifold surfaces can.
2. In the qualitative results of Figure 5, I would also suggest presenting the original images, which helps to determine which results are closer to the ground truth.
3. In Table 2, I would suggest presenting the standard deviation of CD and conducting a statistical significance analysis.
4. The authors only utilize CD for quantitative evaluation, which is not enough to demonstrate the performance of MIND.

---

> ### Author Rebuttal · Authors · 2025-07-30
>
> ## In the qualitative results of Figure 5, I would also suggest presenting the original images, which helps to determine which results are closer to the ground truth.
>
> Thanks for your suggestion. We will provide the original images in the revision.
>
>
>
> ## What is the N.A. in Figure 5?
>
> In this case, N.A. indicates that DMUDF fails to extract any mesh facets from the given UDF.
>
> Specifically, DMUDF uses an octree-based structure to improve computational efficiency, but it relies on the UDF being accurate across the entire spatial domain.
> However, DEUDF, as a UDF learning method, focuses on accuracy near the surface and often produces unstable or noisy UDF values farther away. This inaccuracy leads the DMUDF octree to misidentify valid leaf nodes, resulting in complete reconstruction failure in large regions. As a result, no mesh can be generated for this case, and we mark it as N.A. in Figure 5 to reflect this failure.
>
>
> ## In addition to chamfer distance, are there any other metrics to demonstrate the superiority of MIND?
>
> The main advantage of MIND over previous methods lies in its ability to support non-manifold surface reconstruction. However, there is currently no standardized benchmark for quantitatively evaluating this capability. To partially address this, we incorporate geodesic distance comparisons in our experiments to help assess the quality and correctness of the extracted non-manifold structures.
>
>
>
>
> ## The presented results are obtained on simple scenarios. Can the non-manifold be applied to more realistic and complex scenarios, such as medical image scenarios?
>
>
> Yes, MIND can be applied to more realistic settings. In fact, we have tested our method using UDFs learned by NUDF on the Left Atrial Appendage (LAA) segmentation dataset, which contains anatomically complex structures and potential non-manifold features. We will provide the visual results in the revision.

---

> > ### Comment · Area_Chair_DRw3 · 2025-08-08
> > **Standard deviation of CD and statistical significance analysis**
> >
> > Dear Reviewer Tx5y and Authors,
> >
> > Since Reviewer Tx5y raised further concerns in the "Final Justification", which is not visible to the authors at this moment, I am posting a message here to seek clarification from the authors.
> >
> > Reviewer Tx5y raised the following concern in his review:
> > > In Table 2, I would suggest presenting the standard deviation of CD and conducting a statistical significance analysis.
> >
> > It appears that this point is overlooked or misinterpreted in the rebuttal. Could the authors clarify this?
> >
> > Best,
> > AC

---

> > > ### Author Response · Authors · 2025-08-08
> > >
> > > Dear AC and Reviewer Tx5y
> > >
> > > Thank you for bringing this to our attention. In our revision, we will add the standard deviations and medians of CDs to better analyze the performance of different methods on the datasets.

---

> ### Author Response · Authors · 2025-08-08
> **The Results of The Standard Deviations and Medians**
>
> Thanks for the suggestion from Reviewer Rx5y, we present the results of the standard deviations and medians on the Shapenet-Car dataset and Deepfashion3d dataset. And the following results will be added to the revision.
>
>
> | Standard deviation ($10^{-3}$)     | DMUDF | DCUDF | Ours |
> | ----------- | ----------- | ----------- |----------- |
> |Shapenet-Car| 2.112 | 2.252 | 2.107|
> |Deepfashion3d| 0.512| 0.535| 0.507 |
>
> | Median ($10^{-3}$)     | DMUDF | DCUDF | Ours |
> | ----------- | ----------- | ----------- |----------- |
> |Shapenet-Car| 2.564 | 2.770 | 2.447|
> |Deepfashion3d| 1.383| 1.495| 1.339 |

---

### Official Review · Reviewer_duN2 · 2025-07-02

**Clarity:** 2
**Significance:** 3
**Originality:** 3
**Rating:** 5
**Confidence:** 3

**Summary:**

This paper presents a method for extracting meshes from arbitrary unsigned distance fields, including non-manifold structures. It comprises three stages – firstly a local assignment of 'sides' to near-zero regions of the UDF; then fusion into a single global label field (a material interface) by propagating information outwards from edges; then meshing of the label field. Experiments are conducted on UDFs derived from several applications – from point-clouds, images, and medial fields; these show the proposed method works better than two baselines.

**Questions:**

The bolded claim at L44 seems inconsistent with e.g. DMUDF and NDC – while these do not always work perfectly, they do address the relevant task/setting and can in principle achieve non-manifold surface reconstructions.

Is it possible to perform the qualitative evaluation of similarity of geodesic distance fields instead quantitatively on the shapenet / deepfashion experiments?

See above regarding ablation experiments to justify design decisions – it would be valuable to know the quantitative effect that different aspects of the proposed method have on the final results, in order to help readers judge their importance.

**Ethical Concerns:**

["NO or VERY MINOR ethics concerns only"]

**Final Justification:**

Given the rebuttal responses and discussion, I still favor acceptance. However I am not increasing my rating further, mainly due to the lack of ablations on different aspects of the method (which the authors did not provide even in quantitative form during the rebuttal period), and also the lack of numerical results for topological similarity (which is somewhat important given the focus of the paper on reconstructing specific topologies faithfully).

**Limitations:**

Yes, limitations discussed, though in supplementary. No broader impacts

**Quality:**

3

**Strengths And Weaknesses:**

The paper addresses an interesting problem of UDF reconstruction from points, that has received much less attention than the equivalent for SDFs.

The proposed method is novel. While there are ideas drawn from various other works, overall the approach is quite different in flavor to existing methods solving the task. The different stages of the pipeline are generally well-motivated.

The evaluation is quite comprehensive. Experiments are conducted on UDFs from different real-world pipelines – using two methods for UDF inference from point-clouds, and one for UDF inference from images. Results are significantly better than two other approaches.

Qualitative results exhibit non-manifold edges in reasonable locations while conforming closely to the desired geometry; the result on the shapenet car in fig. 6 is particularly impressive with very fine details of single-sheet geometry correctly recovered.

Quantitative results exceed the performance of the baselines DMUDF and DCUDF, when measuring chamfer distance of reconstructed meshes from a point-cloud, to the original mesh the point-cloud was derived from.

There are no results on variants / ablations of the approach, which would reveal the benefit to the reader of certain design decisions. For example, at L182, small enhancements to M3C are described, which "enhance accuracy" and "further refine", yet there is no experimental validation of the extent to which these help; similarly for the refinement in the paragraph at L185.

The evaluation visualises geodesic distance fields as a way to show the topology of the mesh is similar to a corresponding point-cloud, i.e. that the mesh has not for example 'fattened' surfaces to be watertight. However, this evaluation is only qualitatively, and only shown on two examples. It would be valuable to do this also for the shapenet / deepfashion experiments in Table 2, since chamfer distance is a much poorer measure of topological correctness than similarity of the geodesic distance field. One easy way to achieve this would be to measure mean similarity of geodesic fields at pairs of nearest-points in original and reconstructed meshes.

The subject area is only borderline appropriate for NeurIPS – it is pure computer graphics / geometry processing, with no machine learning involved in the proposed approach (it can be applied to neural UDFs, but that is not in any way intrinsic to the method).

---

> ### Author Rebuttal · Authors · 2025-07-30
>
> ## There are no results on variants / ablations of the approach, which would reveal the benefit to the reader of certain design decisions. For example, at L182, small enhancements to M3C are described, which "enhance accuracy" and "further refine", yet there is no experimental validation of the extent to which these help; similarly for the refinement in the paragraph at L185.
>
>
> Thank you for pointing this out. We agree that ablation studies are valuable for understanding the contribution of each component.
>
> In the revision, we will include additional mesh extraction results that compare our method to the original M3C and our pipeline without DCUDF optimization.
>
> However, due to this year's NeurIPS rebuttal guidelines that prohibit uploading additional figures, we are unable to include these visual results in the current response. We will ensure that these ablations are clearly presented and discussed in the revised submission.
>
>
>
> ## The bolded claim at L44 seems inconsistent with e.g. DMUDF and NDC – while these do not always work perfectly, they do address the relevant task/setting and can in principle achieve non-manifold surface reconstructions.
>
> Thank you for the observation. We agree that both DMUDF and NDC are capable of generating non-manifold surfaces in principle.
> We will revise the statement to clarify our intended meaning: while these methods may produce non-manifold structures, they often do so without explicitly controlling for manifold corrrectness. As a result, they may introduce unintended non-manifold artifacts even in regions that should be manifold.
>
> In contrast, our method is specifically designed to capture non-manifold features where they are required, while preserving the integrity of manifold regions.
>
> We will revise the statement and clarify this distinction in the revision.
>
>
> ## Is it possible to perform the qualitative evaluation of similarity of geodesic distance fields instead quantitatively on the shapenet / deepfashion experiments?
>
> Thank you for the suggestion. ShapeNet could indeed be a suitable candidate for evaluating the effectiveness of MIND. However, in practice, many ShapeNet models are constructed by stacking individually modeled manifold parts, which are not truly non-manifold in a topological sense, and would require extensive data preprocessing and remeshing to convert them into a clean dataset with genuine non-manifold topology.
>
> Once such preprocessing is completed, it would be possible to perform quantitative evaluations, such as comparing geodesic distance fields computed on the extracted surfaces against ground-truth mesh surfaces.
>
> However, because of the significant preprocessing effort required, we consider this to be a direction for future work.
>
>
>
> ## The subject area is only borderline appropriate for NeurIPS – it is pure computer graphics / geometry processing, with no machine learning involved in the proposed approach (it can be applied to neural UDFs, but that is not in any way intrinsic to the method).
>
> Our method is designed to address a critical bottleneck in UDF-based 3D learning pipelines. As an effective mesh extraction algorithm for UDFs, our method directly supports and enhances neural approaches for 3D reconstruction and generation.
>
> UDFs have recently gained significant attention in the AI and NeurIPS communities as a flexible representation for 3D shapes, especially in tasks involving neural implicit functions. Prior works (e.g., NDF, CAP-UDF, NeUDF, NU-NeRF, DreamUDF) demonstrate the advantage of UDFs in representing open surfaces, but their full potential, specifically for representing non-manifold structures, remains underutilized due to the lack of suitable mesh extraction algorithms.
>
> Although some methods such as DMUDF and NDC attempt to extract non-manifold geometry, they often introduce uncontrolled and excessive non-manifold artifacts, even in regions that are intrinsically manifold. In contrast, our method explicitly preserves manifold regions while correctly capturing non-manifold features, unlocking a unique capability of UDFs that was previously inaccessible.
>
> We believe that our method has direct implications for learning-based 3D reconstruction. For example, NU-NeRF reconstructs nested transparent objects by combining multiple SDFs but suffers from discontinuities at non-manifold intersections due to the limitations of SDFs. Similarly, in 3D generation, methods such as TripoSF aim to produce models with rich internal structures, but the use of SDFs prevents mesh reconstruction at non-manifold junctions. Our method may inspire 3D AIGC work on generating non-manifold structures, either by extracting meshes from neural UDFs or, in the future, by attempting to integrate material interface (MI) prediction into neural pipelines.
>
> In summary, we believe our work makes a foundational contribution to UDF-based shape modeling, which is highly relevant to current and emerging research directions in 3D learning. We hope that by presenting this work at NeurIPS, we can stimulate further research in this area and provide a missing tool for the community working on UDF-based 3D reconstruction and generation.

---

> > ### Comment · Reviewer_duN2 · 2025-08-07
> >
> > Thanks for the detailed response. I appreciate the promises to clarify claims etc. Overall I remain broadly positive about the work; still the additional results suggested (quantitative evaluation of topological similarity on remeshed shapenet) plus ablations, would strengthen it. Note for shapenet there exist watertight remeshed versions already, which I think would be suitable for this experiment.

---

> > > ### Author Response · Authors · 2025-08-08
> > >
> > > Thank you for your positive feedback on our work. We agree that a quantitative metric for topology would strengthen the paper.
> > >
> > > The remeshed ShapeNet version we examined (https://github.com/GAP-LAB-CUHK-SZ/RfDNet/tree/main) constructs an SDF for each original model by rendering multi-view images to produce watertight meshes. However, this process removes all internal structures, including any non-manifold geometry, making the dataset unsuitable for evaluating how different algorithms capture non-manifold topologies.
> > >
> > > Similarly, the DeepSDF preprocessing pipeline also prunes faces based on visibility, which eliminates internal structures that may contain non-manifold features.
> > >
> > > Since current ShapeNet remeshing pipelines produce purely manifold meshes and discard non-manifold structures, we cannot use them for our quantitative experiments. A dedicated dataset containing topologically non-manifold meshes would be required for such analysis, but to the best of our knowledge, no such benchmark currently exists.
> > >
> > > Nevertheless, we will expand our test set by collecting more models with genuine non-manifold topologies. We thank the reviewer again for their thoughtful feedback and constructive suggestions.

---

### Official Review · Reviewer_TBUd · 2025-07-03

**Clarity:** 3
**Significance:** 3
**Originality:** 3
**Rating:** 5
**Confidence:** 3

**Summary:**

The authors propose MIND for generating material interfaces directly from unsigned distance fields by deriving a spatial partitioning. A three stage strategy is used - i) construction of a local two-signed distance field, (ii) extending the local two-signed field to a global multi-labeled field, and (iii) extraction of non-manifold surface meshes from the material interface. Qualitative results seem to show strong performance.

**Questions:**

1. How are your input UDF's obtained?
2. Given that you are using voxel-based downsampling of point clouds, can you provide an ablation study of voxel resolution vs accuracy of capturing fine details?
3. the non-manifold surface extraction method mentions using the value of $w_S^l$ to determine the intersection points. Can the authors explain how is this different from Neural Dual Contouring's approach (which uses the corner signs on a grid)?
4. Laplacian smoothing tends to smooth/blur sharp features. Have the authors considered using a quadratic error function optimization to preserve the sharp features?
5. Can the authors provide information on the computational time for MIND and compare it against baselines?
6. Have the authors considered comparing against MeshCAP, MeshUDF?

**Ethical Concerns:**

["NO or VERY MINOR ethics concerns only"]

**Final Justification:**

After reading responses to my questions and other reviewer's questions, I have decided to increase my rating.

**Limitations:**

Yes

**Quality:**

3

**Strengths And Weaknesses:**

strengths:
- Addresses a fundamental gap no other work has proposed before
- Local two-signed field to global multi-labeled field strategy is novel
- Adoption of laplacian smoothing is justified
- Qualitative results indicate strong performance

weaknesses:
- Voxel resolution dependence causes missing out on finer geometric features
- The performance of the method seems to be bottlenecked by the accuracy of the underlying UDF prediction method
- Laplacian smoothing strategy makes it harder to judge the impact of the global labeling aspect of MIND
- Lack of quantitative metrics like resolution sensitivity, parameter sensitivity (r1, r2)

---

> ### Author Rebuttal · Authors · 2025-07-30
>
> ## Voxel resolution dependence causes missing out on finer geometric features
>
> We agree that resolution imposes fundamental limits on geometric accuracy. However, this limitation is not unique to our method, as it is shared by all voxel-based surface extraction techniques, including MC, DC and their many variants.
>
> These methods generate discrete mesh representations, whose level of detail is inherently constrained by the grid resolution. While increasing resolution can recover finer details, the discrete and resolution-dependent nature  is a general characteristic of the entire class of methods.
>
>
>
>
>
> ## The performance of the method seems to be bottle-necked by the accuracy of the underlying UDF prediction method.
>
> This is in fact a common challenge for all mesh extraction algorithms. In general, no surface extraction method can recover geometric details that are not already encoded in the input distance field. However, different methods vary in how well they handle noise or imperfections in the UDF.
>
> Our method demonstrates improved robustness to UDF noise due to the erosion operation, which stabilizes region labeling and removes spurious fragments. Furthermore, the DCUDF-based optimization further refines the extracted geometry, improving accuracy up to the intrinsic limit imposed by the input UDF quality.
>
> Moreover, our method can further improve accuracy without increasing resolution by directly subdividing the mesh extracted from M3C and then performing the optimization process. For example, we trained a UDF on the Armadillo model using DEUDF, achieving a Chamfer Distance (CD) of 0.003484 at a resolution of 256. By applying one subdivision step (splitting each triangle into four), the CD decreases to 0.003335.
>
>
> ## Laplacian smoothing strategy makes it harder to judge the impact of the global labeling aspect of MIND
>
> Thanks for raising this question. It is important to note that the global multi-labeled field in MIND is responsible for determining the topological structure of the reconstructed surface, specifically, how different regions are partitioned and connected.
>
> In contrast, Laplacian smoothing and subsequent DCUDF optimization refine only the geometry of the surface without altering its topology.
>
> The impact of MIND’s global labeling can therefore be assessed through the topology-related results presented in the paper, such as geodesic distance comparisons and highlighted non-manifold edges. These reflect the correctness of the extracted connectivity structure, regardless of geometric smoothing.
>
>
> ## Lack of quantitative metrics like resolution sensitivity, parameter sensitivity (r1, r2)
>
> Thank you for the suggestion.
> The parameters $r_1$ and $r_2$ depend on the grid size and can be flexibly adjusted as long as the conditions provided in the experimental section are met. We will provide quantitative results at different resolutions.
>
>
> ## How are your input UDF's obtained?
>
> Our work focuses on extracting zero level sets from learned UDFs. Several prior works have developed methods to learn UDFs from point clouds or multi-view images. In our experiments, we use UDFs generated by a variety of state-of-the-art learning-based approaches, including CAPUDF, LevelSetUDF, DEUDF, NU-NeRF, and NeUDF. We use them as input to our method to evaluate the effectiveness of MIND in extracting non-manifold surfaces.
>
>
>
>
> ## Given that you are using voxel-based downsampling of point clouds, can you provide an ablation study of voxel resolution vs accuracy of capturing fine details?
>
> Thank you for your suggestion. We use PCA to calculate normals of point clouds. Generally, point cloud orientation algorithms require a dense point cloud to obtain sufficiently reliable normal vectors. Since calculating normals from point clouds and obtaining the local side field is not time-consuming, we do not recommend reducing the resolution of the point cloud downsampling. An excessively low point cloud density will decrease the reliability of normals, thereby generating unnecessary topological structures. We will supplement the relevant experimental results in the revision.
>
>
> ## Can the authors explain how is this different from Neural Dual Contouring's approach (which uses the corner signs on a grid)?
>
> NDC uses a neural network to directly predict the corner signs of grid cells, which are then used to extract surfaces. However, this approach is sensitive to prediction errors due to inherent instability and lack of spatial consistency. For example, NDC may assign inconsistent signs to adjacent cubes where no surface is present, leading to redundant faces in the reconstructed mesh.
>
> In contrast, our method computes the initial local side field using optimization based on geometric cues, rather than learned predictions. This provides enhanced stability and consistency, especially in regions with complex or ambiguous geometry. As a result, our method significantly reduces the types of artifacts commonly observed in NDC.
>
> ## Laplacian smoothing tends to smooth/blur sharp features. Have the authors considered using a quadratic error function optimization to preserve the sharp features?
>
> Thank you for your insightful suggestion. We agree that using the quadratic error function to compute vertex positions is effective for preserving sharp features. However, in this work, we focus on the non-manifold topology identification and non-manifold mesh extraction. Hence, we would like to leave this as a future work to improve the mesh accuracy.
>
>
>
>
> ## Have the authors considered comparing against MeshCAP, MeshUDF?
>
> Both MeshUDF and MeshCAP extend Marching Cubes to support surface extraction from UDFs. However, Marching Cubes is inherently limited to generating manifold structures, and these methods cannot preserve non-manifold structures by design. Since our method specifically targets non-manifold surface reconstruction, we believe it would be inappropriate to perform a direct comparison against approaches that are fundamentally restricted to manifold outputs.
>
> That said, MeshUDF and MeshCAP remain useful baselines for evaluating performance on manifold regions, and we include them where relevant in our experiments. In particular, the original NeUDF employs MeshUDF for mesh extraction, and we reported the NeUDF results in both the main paper and the supplementary material. We will make this connection more explicit in the revised version.

---

> ### Comment · Reviewer_TBUd · 2025-08-06
> **Official comment by Reviewer TBUd**
>
> I appreciate the authors' effort to write the rebuttal and I acknowledge the responses.
>
> I have increased my rating.
>
> Reviewer TBUd

---

> > ### Author Response · Authors · 2025-08-08
> >
> > Thank you for your time and constructive feedback.

---

### Official Review · Reviewer_yUiZ · 2025-07-05

**Clarity:** 3
**Significance:** 3
**Originality:** 3
**Rating:** 4
**Confidence:** 3

**Summary:**

This paper introduces MIND, a new method for extracting non-manifold meshes directly from Unsigned Distance Fields (UDFs). Unlike traditional approaches that reconstruct Signed Distance Fields (SDFs) locally—often resulting in topological artifacts or inability to handle non-manifold geometries—MIND derives a global multi-labeled field to identify material interfaces. This enables accurate surface extraction via a multi-labeled Marching Cubes algorithm, effectively managing complex non-manifold surfaces across diverse data sources, including point clouds and multi-view reconstructions. The approach outperforms existing methods in robustness and accuracy.

**Questions:**

Why in Eq. 1 and 2 are the indicator functions normalized by a cubic term? It would be really good to have a limitations section in the paper. The MI insight seems cool, but is it always applicable. Further insights into where it works and where it fails would be invaluable.

**Ethical Concerns:**

["NO or VERY MINOR ethics concerns only"]

**Limitations:**

Yes

**Paper Formatting Concerns:**

None.

**Quality:**

3

**Strengths And Weaknesses:**

The promise of UDFs have been well understood for some time, and the authors do a good job of walking the reader through the drawbacks of current approaches for extracting surfaces – in particular the sensitivity of Marching Cubes. The authors claim that at the time of publication, there is no method that can effectively extract non-manifold surfaces directly from a UDF. The central idea of the paper is to employ material interfaces (MIs), which is essentially a partitioning of the spatial domain into multiple labelled regions. What makes this work novel, is that they extend the concept of MIs to support both closed and open manifolds. They do this by introducing an additional label category for background material excluding all faces adjacent to it. The proposed approach offers impressive performance against DMUDF [22] and DCUDF [24]. The surface extraction results on the ShapeNet-Car dataset in Fig. 6 are impressive. The advantage of their approach over techniques like NU-NERF [41] is compelling.

---

> ### Author Rebuttal · Authors · 2025-07-30
>
> ## Why in Eq.1 and 2 are the indicator functions normalized by a cubic term?
>
> The sign of a query point in our local side field is determined based on the normal vectors of nearby surface points. In manifold regions, these normals are typically consistent, leading to a stable sign determination. However, near non-manifold structures, multiple surface patches with differing normals may intersect. This can make the sign assignment ambiguous or unstable.
>
> To address this, we adopt a formulation inspired by the generalized winding number, where the contribution of each neighboring point is inversely weighted by the cube of its distance to the query point. This cubic normalization gives more influence to points that are closer, effectively biasing the sign toward the dominant local geometry. In particular, it encourages alignment with the normal vector of the nearest surface patch, helping to suppress fluctuations caused by distant conflicting normals.
>
> This weighting strategy improves robustness and locality in sign estimation, especially near regions with complex topology or non-manifold junctions.
>
> ## It would be really good to have a limitations section in the paper. The MI insight seems cool, but is it always applicable. Further insights into where it works and where it fails would be invaluable.
>
> Thanks for your suggestions. Our method can reliably generate non-manifold features where needed, in contrast to dual contouring variants that may introduce spurious non-manifold elements even in manifold regions. However, some limitations remain.
>
> First, our approach requires an erosion operation, which may inadvertently remove small geometric structures and lead to missing facets. We generally advise against using excessively low resolutions during erosion. If a lower resolution is necessary elsewhere (e.g., for sampling the unsigned distance field), we recommend applying erosion after upsampling the data.
>
> Second, due to the optimization step involved in our pipeline, the method is more computationally expensive than non-optimization-based approaches. Nevertheless, this additional cost is justified by the significant improvement in reconstruction quality—particularly in faithfully preserving non-manifold structures, which existing methods are unable to achieve.
>
> Please also see the discussion on limitations in the supplementary material.

---

> > ### Comment · Reviewer_yUiZ · 2025-08-08
> >
> > Thank you for your detailed and thoughtful rebuttal. I appreciate the clarifications and revisions you have provided. I am satisfied that my earlier concerns have been adequately addressed, and I have no further substantive comments.

---

> > > ### Author Response · Authors · 2025-08-08
> > >
> > > Thank you for your time and constructive feedback.

---

### Note · Authors · 2025-08-12

Dear Area Chairs and Reviewers,

We thank you for your constructive feedback, which has helped strengthen our work. Our method addresses a critical, previously unexplored challenge in UDF-based non-manifold mesh extraction, advancing both 3D reconstruction and 3D deep learning. Below, we summarize our main contributions:

**Novelty & Significance**: Address a fundamental, unprecedented gap in UDF-based non-manifold mesh extraction.

**Unique Approach**: Propose a novel approach to identify distinct multi-labeled fields and generate material interfaces from UDF.

**Robustness & Accuracy**: Impressive performance against existing methods (DMUDF, DCUDF) in complex non-manifold scenarios, with superior detail recovery in qualitative results.

**Justified Methodology**: Logical, well-motivated three-stage pipeline, including Laplacian smoothing.

**Comprehensive Evaluation**: Extensive experiments across diverse data sources validate effectiveness.

We appreciate the ACs’ and reviewers’ feedback, which has further improved this work. We are confident that MIND makes a meaningful contribution to non-manifold surface reconstruction from UDFs and will benefit the NeurIPS and 3D deep learning communities.

---

### Decision · Program_Chairs · 2025-09-17

**Decision:**

Accept (poster)

**Comment:**

This paper proposes a novel method for extracting non-manifold meshes directly from Unsigned Distance Fields (UDFs).

The paper initially received one accept and three borderline-accept scores. Reviewers highlighted several strengths:
- Extending material interfaces to support both closed and open manifolds is a novel idea.
- Evaluation is comprehensive.
- Results demonstrate strong performance.

There were some concerns/questions raised, including the lack of an ablation study, clarification of differences from existing methods, limitations, and applicability to more complex scenarios. The authors addressed most of these concerns in their rebuttal. The final ratings are 2× accept, 1× borderline accept, and 1× borderline reject. Reviewer Tx5y suggested borderline reject due to the absence of standard deviation of CD and a statistical significance analysis, and these were provided during the rebuttal. Given this, the AC believes the merits of the work warrant acceptance. The authors are strongly encouraged to incorporate the promised updates in the final version.